# Vertical Federated Unlearning on the Logistic Regression Model

Zihao Deng [1], Zhaoyang Han [1], Chuan Ma [2], Ming Ding [3], Long Yuan [4], Chunpeng Ge [5] and Zhe Liu [2,*]

[1] School of Computer Science and Technology, Nanjing University of Aeronautics and Astronautics, Jiang Ning District, Nanjing 210000, China; zihao_deng@nuaa.edu.cn (Z.D.); sunrisehan@nuaa.edu.cn (Z.H.)
[2] Zhe Jiang Lab, Kechuang Avenue, Yuhang District, Hangzhou 310000, China; chuan.ma@zhejianglab.edu.cn
[3] Data61, CSIRO, Sydney, NSW 2770, Australia; ming.ding@data61.csiro.au
[4] School of Computer Science and Technology, Nanjing University of Science and Technology, Xiao Lingwei Street, Xuan Wu District, Nanjing 210000, China; longyuan@njust.edu.cn
[5] Joint SDU-NTU Centre for Artificial Intelligence Research (C-FAIR) & Software School, Shandong University, Jinan 250000, China; chunpeng_ge@sdu.edu.cn
* Correspondence: zhe.liu@zhejianglab.com

**Abstract:** Vertical federated learning is designed to protect user privacy by building local models over disparate datasets and transferring intermediate parameters without directly revealing the underlying data. However, the intermediate parameters uploaded by participants may memorize information about the training data. With the recent legislation on the "right to be forgotten", it is crucial for vertical federated learning systems to have the ability to forget or remove previous training information of any client. For the first time, this work fills in this research gap by proposing a vertical federated unlearning method on logistic regression model. The proposed method is achieved by imposing constraints on intermediate parameters during the training process and then subtracting target client updates from the global model. The proposed method boasts the advantages that it does not need any new clients for training and requires only one extra round of updates to recover the performance of the previous model. Moreover, data-poisoning attacks are introduced to evaluate the effectiveness of the unlearning process. The effectiveness of the method is demonstrated through experiments conducted on four benchmark datasets. Compared to the conventional unlearning by retraining from scratch, the proposed unlearning method has a negligible decrease in accuracy but can improve training efficiency by over 400%.

**Keywords:** federated unlearning; machine unlearning; vertical federated logistic regression

## 1. Introduction

The success of deep learning is largely attributed to training on large-scale datasets. However, traditional training methods require centralizing data in one host, which can present challenges when dealing with sensitive or private participant data. Additionally, some participants may not want to share their data with others, making traditional methods impractical. To address these data silo and privacy issues, researchers have proposed federated learning (FL) [1]. FL enables participants to train a global model based on locally held private data and exchange intermediate parameter updates. There are two main types of FL: horizontal FL (HFL), in which parties share the same features but hold different samples, and vertical FL (VFL), in which participants share the same samples but hold different sets of features [2]. Although FL has shown promise in addressing data privacy concerns, it is not without its challenges. One such challenge is the need to "unlearn" previously learned data without data breaches or other privacy violations. Although unlearning has been studied extensively in centralized learning, it is a relatively new area of research in the context of FL. In this paper, the potential of federated unlearning is explored, and a novel algorithm is proposed for effectively unlearning the vertical federated logistic regression model.

Unlearning is the process of removing previously learned data from a model without compromising its performance on the remaining data. From a privacy perspective, recent legislation has emphasized the "right to be forgotten" [3], such as the General Data Protection Regulation (GDPR) in European [4] and the California Consumer Privacy Act (CCPA) in the United States [5]. They claim that participants have the right to request that the global model should have the ability to unconditionally forget their private data information when the machine-learning task is completed. This is particularly important in FL, where multiple parties may be contributing data to the model, and one party may need to remove its data due to a privacy violation or other issue. In addition to the "right to be forgotten", forgetting the contribution of the client's private data is also beneficial. Considering that FL is vulnerable to data-poisoning attacks [6,7], eliminating the influences of these data can greatly improve the security, reliability, and robustness of FL systems.

Since the global model may potentially contain private participant data, directly deleting participants' private data has proven to be infeasible [8]. One naive approach to make the global model forget the private data of a participant is to retrain the model from scratch, but this method comes with significant computational and communication costs. Moreover, FL is an incremental learning process [9], and each round of global model updates relies on previous updates from participants. If the contribution of one participant is removed from the aggregation process, the global model received by other participants and subsequent updates based on the global model will become invalid. As a result, implementing federated unlearning is an extremely difficult task. Existing research on federated unlearning mainly focuses on HFL [10,11], and there is no research on federated unlearning in VFL. In addition to achieving unlearning, improving efficiency is also a significant challenge. Existing federated unlearning methods require retraining the unlearned model for multiple rounds to recover the model's prediction accuracy after forgetting the private data of participants.

The problem of federated unlearning in VFL is addressed in this paper, where the forgetting of a specific participant's private data is equivalent to the forgetting of certain features in the federated dataset. A novel method is proposed that minimizes the sum of all participants' parameters while imposing constraints on each of them, enabling the quick convergence of the unlearned model after forgetting a specific client's contribution. Furthermore, the effectiveness of the method against data-poisoning attacks is evaluated [12]. Experiments are conducted on four benchmark datasets to demonstrate the effectiveness of the method, and the results indicate significant improvements in terms of unlearning and convergence speed.

The major contribution of this paper can be summarized as follows:

- A vertical federated unlearning method is proposed in this paper by imposing constraints on intermediate parameters and subtracting target client updates from the global model. The unlearned model can converge quickly and efficiently after forgetting the private data of the target client. This is the first efficient unlearning algorithm in VFL that can achieve efficient unlearning on the logistic regression model.
- Data-poisoning attacks were implemented to evaluate the effectiveness of the federated unlearning method in VFL. In this study, a scenario was simulated where a target client participated in the model training process with toxic data. Subsequently, the ability of the unlearned model to eliminate the impact of these attacks after removing the client's contribution was tested. This is an important evaluation metric because it shows that the unlearning method not only removes the target client's private data but also eliminates any adverse effects caused by data-poisoning attacks from that client.
- Empirical studies were conducted on four real-world datasets to evaluate the performance of the proposed method. The results show that the proposed unlearning method can efficiently remove the influence of the target client's private data and recover the accuracy of the global model with only one extra round of updates. This is a significant improvement in efficiency compared to retraining from scratch, which can be computationally expensive and time-consuming.

This paper begins with an introduction to the background and motivation of vertical federated unlearning, which aims to protect user privacy data from model leakage. Next, an unlearning method is proposed to address these challenges. It involves imposing constraints on the uploaded intermediate parameters during the training phase and removing the target client during the unlearning phase. Furthermore, the other clients are updated using the intermediate parameters uploaded by the target client in the last round. In the subsequent section, ablation experiments were performed on four benchmark datasets to assess the efficacy of the unlearning method. The evaluation was based on the analysis of prediction accuracy and success rate of poisoning attacks on the unlearned model. Subsequently, a summary of the contributions, limitations of the proposed method, and potential avenues for future research were provided.

## 2. Related Work

Several previous studies have addressed the problem of unlearning in various machine-learning settings. Cao and Yang introduced the concept of machine unlearning and proposed an efficient unlearning algorithm based on transforming the optimization objective function, which enables the removal of a training data sample by updating only a small fraction of summation, resulting in faster unlearning than retraining from scratch [13]. Ginart and Guan proposed two efficient unlearning methods for the k-means clustering algorithm, which make use of model compression techniques to remove the effect of specific data samples from the clustering model [14]. Bourtoule et al. introduced the SISA framework to reduce time overhead by increasing space overhead. The framework divides the dataset into shards and slices and removes the training data samples from the shards and slices during unlearning, after which the corresponding shards and slices are retrained [9]. Sekhari et al. proposed an unlearning method that utilizes the Hessian matrix to identify an approximately optimal solution for a strongly convex problem. Then, the solution is perturbed to introduce some level of uncertainty to the optimizer within a small radius, resulting in a fuzzy removal of the target data from the model [15]. Gupta et al. proposed a method that applies differential privacy to adaptive training, which reduces the deletion guarantees for adaptive sequences to those for non-adaptive sequences, thus achieving more efficient unlearning [16]. Tarun et al. introduced a novel machine unlearning framework with error-maximizing noise generation [17] and impair-repair-based weight manipulation that offers an efficient solution to the above questions. Chundawat introduced the novel problem of zero-shot machine unlearning [18] that caters to the extreme but practical scenario where zero original data samples are available for use. However, these existing machine unlearning algorithms are not suitable for vertical federated unlearning scenarios. This is because vertical federated unlearning focuses on forgetting user data features in the model, rather than simply removing data samples. Therefore, a different approach is needed to effectively address the challenges of vertical federated unlearning.

In the context of FL, Gong et al. proposed the FORGET-SVGD algorithm, a federated unlearning method for Bayesian models [19], which improves unlearning performance with the use of SVGD. Liu et al. tackled the problem of unlearning in FL by training the model backward using the gradient ascent method [20], but their method increases communication overhead. Wu et al. significantly improved the training speed of federated unlearning using knowledge distillation after forgetting the target client data [21]. Wei et al. proposed a user-level privacy protection scheme based on differential privacy technology [22]. Yuan et al. proposed an efficient unlearning method FRU for recommendation systems [23], inspired by the log-based rollback mechanism of transactions in database management systems. Wang et al. proposed new taxonomies to categorize and summarize the state-of-the-art federated unlearning algorithms [24] and summarize defense techniques with the potential of preventing information leakage. Zhu et al. proposed FedLU, a novel FL framework for heterogeneous knowledge graph embedding learning and unlearning [25]. Moreover, they present an unlearning method based on cognitive neuroscience. Li et al. proposed a simple-yet-effective subspace-based federated unlearning method, dubbed SFU [26], that

lets the global model perform gradient ascent in the orthogonal space of input gradient spaces formed by other clients to eliminate the target client's contribution without requiring additional storage. However, previous work in HFL required retraining the model multiple rounds to recover the model accuracy. In this paper, an unlearning algorithm is proposed in VFL that requires only one extra round of updates to recover the accuracy, demonstrating a significant improvement in efficiency.

To assess the effectiveness of the unlearned model, a data-poisoning attack method specifically designed for VFL was devised. It is worth noting that existing research on data-poisoning attacks primarily focuses on HFL settings. Tolpegin et al. demonstrated that data-poisoning attacks can cause substantial drops in classification accuracy and recall [7], even with a small percentage of malicious participants. Moreover, they propose a defense strategy that can help identify malicious participants in FL to circumvent poisoning attacks. Fang et al. proposed a poison attack method against the local model in FL [27], where the attack effect is achieved by updating the local model parameters in the opposite direction. Fung et al. conducted an evaluation of the vulnerability of federated learning to Sybil-based poisoning attacks [28]. They proposed a defense method called FoolsGold, which outperforms existing state-of-the-art approaches in countering Sybil-based label-flipping and backdoor-poisoning attacks. Zhou et al. proposed a novel optimization-based model poisoning attack that emphasizes the effectiveness, persistence, and concealment of attacks [29]. Sun et al. proposed a client-based defense, named FL-WBC [30], which can mitigate model poisoning attacks that have already polluted the global model. The key idea of FL-WBC is to identify the parameter space where a long-lasting attack effect on parameters resides and perturbs that space during local training. Chen et al. proposed H-CARS [31], a novel strategy to poison recommender systems via CFs. By reversing the learning process of the recommendation model, this method can generate fabricated user-profiles and their associated interaction records for the aforementioned surrogate model. However, existing research primarily concentrates on data-poisoning attacks in HFL settings. In this study, the attack techniques were extended and adapted to the VFL context to evaluate the effectiveness of the unlearned model.

## 3. Problem Definition

As privacy protection regulations such as the GDPR and CCPA become more stringent, individuals have the right to demand that systems delete their private data within a reasonable time. However, this poses a challenge for existing FL systems as they must remove the impact of the private data of participants from the global model. This paper focuses on the Vertical Federated Learning (VFL) system and defines the objective of unlearning in this scenario. The challenges of implementing unlearning in VFL systems are also highlighted. Furthermore, the effectiveness of unlearning in the VFL system is evaluated through the introduction of data-poisoning attacks.

### 3.1. Vertical Federated Logistic Regression

In VFL, $N$ clients collaborate to train a machine-learning model while keeping their private datasets local. In contrast to HFL, VFL participants share the same sample space but different feature spaces. Each participant has a regional model, which they train with private data to obtain intermediate parameters in each training round. Participants then upload their intermediate parameters to the server, and the server returns the aggregated intermediate parameters to each participant. In vertical federated logistic regression, participants update their local model with the following cross-entropy loss function:

$$\ell(w_i; x_i, y_i) = -\frac{1}{N} \sum_{i=1}^{N} [y_i log(f(\sum_{j=1}^{M} w_j x_{ij})) + (1 - y_i) log(1 - f(\sum_{j=1}^{M} w_j x_{ij}))], \tag{1}$$

where $N$ denotes the number of samples in the dataset and $M$ denotes the number of clients, $w_j$ denotes VFL model parameters belonging to the client $C_j$. Moreover, $x_{ij}$ denotes the $i$-th

private sample feature of client $C_j$, $y_i$ denotes the $i$-th private sample true label, and $f(\cdot)$ represents the normalization function, which is usually used in vertical federated logistic regression as sigmoid function.

In the inference process of VFL, participants collaborate with their local models to make predictions. Each participant uses their private data and local model to predict their samples. Then, they send the prediction results to the server for aggregation. The server aggregates the results uploaded by each participant to obtain the final prediction results. The aggregation method of the inference stage is as follows:

$$\hat{y} = f(\sum_{j=1}^{M} w_j x_j), \tag{2}$$

where $M$ denotes the number of clients, $w_j$ and $x_j$ denotes VFL model parameters belonging to the client $C_j$, and $f(\cdot)$ represents the normalization function.

### 3.2. Federated Unlearning

Federated unlearning is the process of eliminating the effect of a specific client's update on the global model in a collaborative learning setting. The goal is to create a new global model that excludes the contribution of the target client, therefore ensuring that the model remains accurate and unbiased. This means that after $N$ clients have collaboratively trained an FL global model, the contribution of the target client is eliminated from the global model. A new unlearned global model is created, which is similar to the model obtained from the training of other $N - 1$ clients.

This task is particularly relevant for vertical federated learning, where each client holds private data and has a local model for collaborative training and prediction. In this context, unlearning a specific client's data can be thought of as removing its features from the federated datasets. However, horizontal federated unlearning methods cannot be directly applied to vertical federated unlearning, as the latter involves removing data samples rather than data features.

The unlearning problem in vertical federated logistic regression is investigated in this paper. It is considered that the sum of the intermediate parameters uploaded by clients is proportional to the gradient of model parameters in the vertical federated logistic regression. Therefore, the proposed method of achieving vertical federated unlearning is by imposing constraints on intermediate parameters during the training process and then subtracting the target client updates from the global model. The advantages are that this method does not need any other clients for training and only one extra round of updates to recover the performance of the previous model.

### 3.3. Evaluation of Vertical Federated Unlearning

Backdoor attacks have been commonly used to evaluate the unlearning effects in HFL [32]. However, such attacks are not effective on table data, which is typically used in VFL. Therefore, it is necessary to develop a corresponding evaluation method that is tailored to table data in VFL. To tackle this challenge, the evaluation of unlearning effects is conducted using data-poisoning attacks. Data-poisoning attacks are one of the most powerful attacks on VFL systems, as they can induce models to classify specific samples incorrectly. These attacks include clean-label poisoning attacks and label-flipping attacks [33]. Label-flipping attacks are the main focus of the experiments, where a portion of the labels is flipped from one class to another while keeping the data features unchanged. This property makes label-flipping attacks a well-established method for evaluating the effectiveness of unlearning.

Specifically, in the evaluation of vertical federated unlearning, It is assumed during the training process, the target client holds some characteristic columns of the dataset and has access to the dataset labels. The target client then intentionally modifies a portion of the labels in the dataset to create toxic samples, which are then mixed with the normal samples

and used for collaborative training. In this way, the trained VFL global model becomes sensitive to the toxic samples used in training, while its prediction success rate for normal samples remains as unchanged as possible. The effectiveness of the unlearning method is evaluated by testing the success rate of data-poisoning attacks on the trained model. In the experiments, the success rate of data-poisoning attacks was calculated as follows:

$$P_{attack} = \frac{count(M(\tilde{X}) == \tilde{Y})}{count(\tilde{D})},$$

(3)

where $\tilde{D}$ represents the number of toxic samples, $M(\tilde{X})$ represents the prediction of the global model for toxic sample $\tilde{X}$, and $\tilde{Y}$ represents the label of the toxic sample.

The effectiveness of the proposed unlearning method is evaluated by testing the success rate of data-poisoning attacks on both the trained global model and the unlearned model. A high success rate of data-poisoning attacks on the trained global model indicates that it is sensitive to the toxic samples used in training. Conversely, a low success rate of data-poisoning attacks on the unlearned model suggests that the proposed unlearning method effectively eliminates the effect of data-poisoning attacks on the target client. Therefore, the unlearned model is expected to eliminate this effect as well. The assumption in this study is that all clients participating in the task possess clean data labels. To assess the unlearning effect after forgetting the target client, an arbitrary client was designated to provide clean data labels, while the labels held by other clients are excluded from the vertical federated learning training process.

### 3.4. Challenges in Vertical Federated Unlearning

#### 3.4.1. Unlearning Sample Features

In VFL, participants have the same user space but different feature space. Forgetting the contribution of the target client is equivalent to forgetting the features of the target sample from the global sample feature space. This differs from HFL, where the sample space of clients is different, and the target is forgetting samples. Therefore, existing unlearning algorithms in HFL, such as Liu's "FEDERASER" [20] and Gong's "FORGET-SVGD" [19], are invalid in vertical federated unlearning. Researchers are required to propose unlearning algorithms that can forget target sample features.

#### 3.4.2. Restrictions on Access to Datasets

After training a VFL task, participants may have deleted the training data related to that task. However, whenever one participant deletes the training data related to the task, the training data stored by other parties all become invalid [9]. As a result, it may be impossible to obtain the complete training datasets for the VFL task. This makes the method of training a model from scratch to unlearning become infeasible since the client may not store the training data for the task.

#### 3.4.3. Limited Information Sharing

In VFL, each client only shares a small amount of information with other clients, which makes it challenging to achieve efficient unlearning. Unlearning in VFL requires that the target client's contribution to the global model be erased while retaining the contributions of other clients. However, due to the limited information sharing, it is difficult to determine which parameters are related to the target client's contribution and which parameters are related to other clients' contributions. Therefore, researchers need to develop new unlearning methods that can effectively distinguish the contributions of different clients based on limited information sharing.

#### 3.4.4. Incremental Training Process

The training of a VFL model is an incremental process, with each round of updates depending on all previous rounds of updates. If the intermediate parameters uploaded

by a client during a particular round of aggregation are modified, the global model after aggregation will change, and all subsequent updates by all clients based on that model become invalid.

## 4. Unlearning Method

To tackle the challenges in vertical federated unlearning outlined in Section 3.4, A new federated unlearning algorithm is proposed in this study, as shown in Algorithm 1. By imposing constraints on the intermediate parameters during the training process of VFL, this method minimizes the sum of the intermediate parameters uploaded by each party, which ensures a speedy convergence of the unlearned model. In the unlearning process, the server removes the local model of the target client from the VFL global model and corrects the global model to obtain the unlearned model. This method can exactly eliminate the contribution of the target client and significantly reduce the learning overhead in federated unlearning.

---

**Algorithm 1** Vertical Federated Unlearning

---

**Input**: Number of clients $N$, Datasets $X_i$, Target client $t$
**Parameter**: Training epoch $K$, Constraint factors $\lambda_i$, learning rates $\theta_i$
**Output**: The unlearned model $\theta_i'$

1: **for** $j = 1, 2, \ldots, K$ **do**
2:     **for** each party $i = 1, 2, \ldots, N$ in parallel **do**
3:         Party $i$ computes $H_i = G_i(x_i, \theta_i)$;
4:         Party $i$ sends $H_i$ to Server s;
5:     **end for**
6:     Server $s$ aggregate $H = F(H_i)$;
7:     Server $s$ sends $H$ to all Parties;
8:     **if** $j = K$ **then**
9:         Server $s$ store last round of $H_i$;
10:     **end if**
11:     **for** each party $i = 1, 2, \ldots, N$ in parallel **do**
12:         Party $i$ computes gradients $\nabla L(\theta_i^j)$ with $H$;
13:         Party $i$ updates $\theta_i^{j+1} = \theta_i^j - \eta_i \nabla L(\theta_i^j) - \lambda_i H_i$;
14:     **end for**
15: **end for**
16: Server $s$ send $-H_t$ to all clients besides target client $t$;
17: **for** each party $i = 1, 2, \ldots, N$ except $t$ in parallel **do**
18:     Party $i$ computes gradients $\nabla L(\theta_i^K)$ with $-H_t$;
19:     Party $i$ updates $\theta_i' = \theta_i^K - \eta_i \nabla L(\theta_i^K) + \lambda_t H_t$;
20: **end for**
21: **return** unlearned model $\theta_i'$;

---

The flowchart of the vertical federated unlearning algorithm is presented in Figure 1. The algorithm takes as input the local dataset of the encountered client and the target client. During the training process, all clients utilize loss training models with intermediate parameter constraints. In the unlearning process, the server initiates the removal of the target client from the global model. Subsequently, the server proceeds to update the other clients with the intermediate parameters uploaded by the target client. Finally, the unlearned model is obtained, completing the unlearning process. The flowchart provides a visual representation of the algorithm's steps and helps illustrate the overall workflow of vertical federated unlearning.

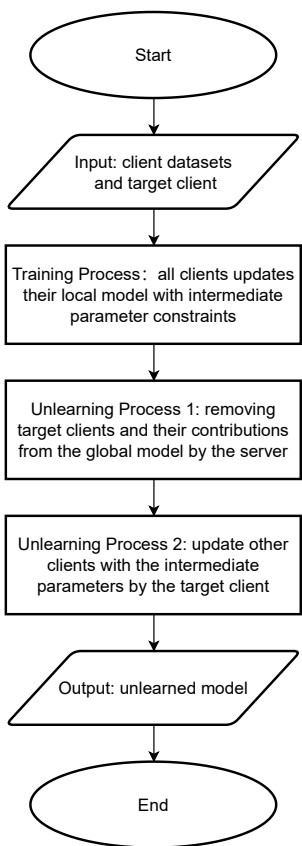

**Figure 1.** Vertical federated unearning algorithm flowchart.

*4.1. Training Process*

In the VFL training process, let $N$ clients train a vertical federated logistic regression Model $M$, where each client $C_i$ has its own local Model $M_i$ and datasets $X_i$, and only one party has the label $Y$. The vertical federated logistic regression model is trained using the loss function Equation (1), and the gradient of the client parameter $W_i$ can be calculated as shown in Equation (4):

$$G_i = (Y - f(\sum_{j=1}^{N} W_j X_j))X_i. \tag{4}$$

To protect the participant's data privacy, the training data are stored locally on the client. To calculate the gradient $G_i$ in Equation (4), clients need to upload intermediate parameters $H_i$ for aggregation. The intermediate parameters $H_i$ to be uploaded by each client are as shown in Equation (5):

$$H_i = W_i X_i. \tag{5}$$

The server aggregates the intermediate parameters uploaded by each client and returns the aggregated results to each client. The clients use the aggregated results to update their respective local model parameters $W_i$.

The VFL model training method is highly efficient and effective, but it poses a significant challenge in performing federated unlearning to remove the target client's contribution. During inference, all clients obtain the intermediate value of the inference based on their locally held sample features and local model. The server aggregates the intermediate value of the prediction from all client local models to obtain an accurate prediction. However, after removing the target client's contribution, the prediction obtained by aggregating the remaining clients' local models may change drastically, leading to a significant impact on the prediction accuracy of the unlearned model. To restore the prediction accuracy of the unlearned model, a substantial computational and communication overhead is neces-

sary. Hence, it is crucial to improve the prediction accuracy of the unlearned model after removing the target client.

It is considered that the update gradient of client parameter $G_i$ should be approximately zero when the model is trained to converge. In Equation (4), the data feature $X_i$ is fixed, so the difference between the data label and the sum of the intermediate parameters uploaded by each client should be approximately zero. However, this constraint only applies to the sum of the intermediate parameters uploaded by all clients, without restricting the intermediate parameters uploaded by individual clients. A client may upload intermediate parameters with large absolute values and still satisfy the condition that the data label minus the sum of the intermediate parameters is zero. However, this can cause the sum of the intermediate parameters of other clients to change dramatically after the target client's contribution is forgotten. Recovering the prediction accuracy of the unlearned model in vertical federated unlearning requires substantial computational and communication overhead. Therefore, the addition of constraints is proposed to minimize the intermediate parameters uploaded by each client in the training process of the VFL model. The loss function with the intermediate parameter constraint added is shown in Equation (6):

$$\ell(w_i; x_i, y_i) = -\frac{1}{N} \sum_{i=1}^{N} [y_i log(f(x_i) + (1 - y_i)log(1 - f(x_i))] - \lambda_i H_i^2, \qquad (6)$$

where $-\frac{1}{N} \sum_{i=1}^{N} [y_i log(f(x_i) + (1 - y_i)log(1 - f(x_i))]$ is the original cross-entropy loss function for vertical federated logistic regression training. $\lambda_i H_i$ is a constraint imposed on the intermediate parameters uploaded by each client, which is used to minimize the intermediate parameters in the unlearning process.

Adding regularization constraints to the intermediate parameters uploaded by all clients can help minimize the sum of the intermediate parameters, resulting in a smaller change in the sum of the intermediate parameters of the remaining clients after removing the target client. This, in turn, can lead to a parameter gradient of the unlearned model that is closer to zero and requires less overhead to restore the prediction accuracy of the unlearned model.

In addition, to correct the global model after forgetting the contribution of the target client, the server needs to store the intermediate parameters uploaded by each client in the previous round.

### 4.2. Unlearning Process

Suppose that $N$ clients cooperate to train a vertical federated logistic regression model, resulting in a global model of $M$ with local model parameters $W_i$ for each party. After training is completed, client $C_i$ wishes to forget its contribution to the global model. If the target client $C_i$ is completely unlearned and the global Model $M$ is corrected accordingly, the unlearned Model $M'$ is obtained as shown in Equation (7):

$$Y = f(\sum_{j=1}^{i-1}(W_j'X_j + b_j') + \sum_{j=i+1}^{N}(W_j'X_j + b_j')). \qquad (7)$$

Since VFL requires all participants to make predictions collaboratively in the prediction process, in vertical federated unlearning, It is necessary to ensure that the predictions made by the unlearned Model $M'$ are as close as possible to the predictions made by the model retrained with other $N-1$ clients. The VFL model collaboration makes predictions as shown in Equation (8):

$$Y = f(\sum_{j=1}^{i-1}(W_jX_j + b_j) + \sum_{j=i+1}^{N}(W_jX_j + b_j) + (W_iX_i + b_i)). \qquad (8)$$

Our objective is to obtain the model of Equation (7) from the model of Equation (8) without training with the client datasets. It is considered that all clients share the same

sample spaces and different feature spaces in VFL. Forgetting the contribution of the target client is equivalent to forgetting the influence of the feature space owned by the target client, which is equivalent to removing the effect of the target client's local model parameters $W_i$ on the global model. Therefore, in the unlearning process, the server is required to remove the local model of the target client $C_i$ from the global model and correct the unlearned model of the other $N-1$ clients.

To correct the unlearned model, the server needs to store the intermediate parameters uploaded by each client in the previous round during the global model training. When the contribution of the target client is to be forgotten, the server retrieves the intermediate parameters uploaded by the target client in the last round and negates them to obtain the aggregated intermediate parameters. Each client then updates their local model parameters using this intermediate parameter. The update rule for the client-side model parameters is given by Equation (9):

$$W_i' = W_i^K - \eta_i \nabla L(W_i^K) + \lambda_t H_t, \tag{9}$$

where $-H_t$ represents the last round of intermediate parameters uploaded by the target client, and $\nabla L(W_i^K)$ is the gradient computed by the unlearned model using $-H_t$.

### 4.3. Feasibility Study

In this subsection, a detailed explanation will be provided on how the server can utilize the negative intermediate parameters from the last round of the target client to correct the unlearned model. It is assumed that clients $C_A$, $C_B$, and $C_C$ cooperate to train a VFL model, and the VFL global model converges after $T$ rounds of training. As mentioned earlier, when the model obtained through VFL training converges, the model parameter gradient should be approximately zero, and the sum of the intermediate parameters uploaded by all participants should also be approximately zero. At this point, the intermediate parameters uploaded by each client are shown in Figure 2.

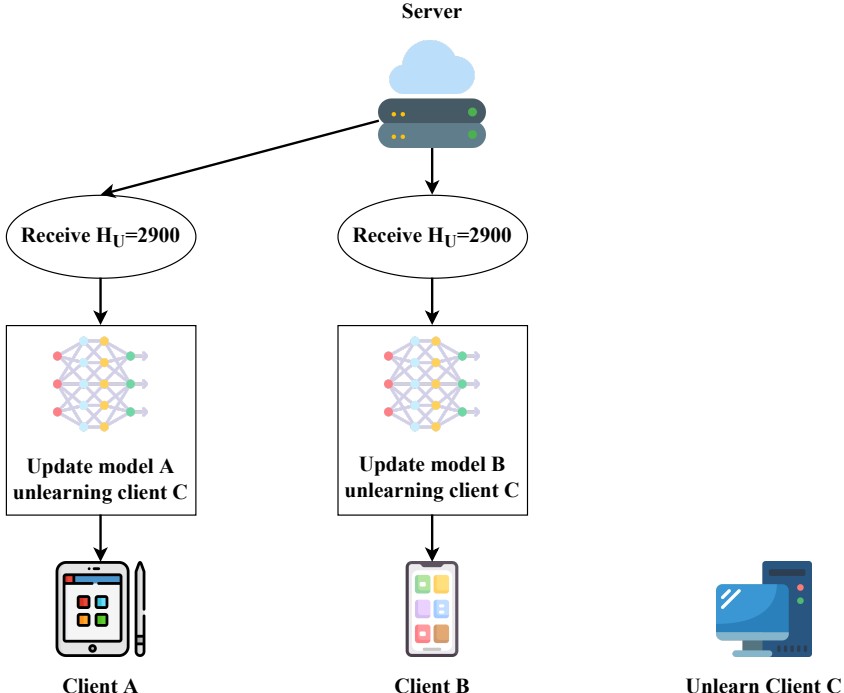

**Figure 2.** Server send unlearning intermediate parameters $H_u$.

Based on the previous arguments, in VFL scenarios, federated unlearning forgets the contribution of the target client's sample features, which are represented by their local model parameters. After removing the contribution of the target client, the VFL global

model eliminates the corresponding local model parameters. However, the sum of the intermediate parameters uploaded by the remaining clients is no longer approximately zero and may change significantly.

Therefore, to forget the contribution of client $C$ from the VFL global model, the local model of client $C$ needs to be removed and then correct the local models of the other clients. It can be observed that the sum of the intermediate parameters uploaded by the clients in Figure 2 is $H_A + H_B + H_C = 100$. If the local model of client $C$ is removed, the aggregated intermediate parameters uploaded by clients $A$ and $B$ should be $H_A + H_B = 3000$. However, the sum of intermediate parameters uploaded by clients should converge to zero when the VFL global model converges. Therefore, after removing the local model of target client $C$ and correcting the unlearned model, the sum of intermediate parameters uploaded by clients $A$ and $B$ should converge to zero.

To achieve convergence of the unlearned model, it is necessary to reduce the intermediate parameters uploaded by clients $C_A$ and $C_B$ to near zero. As previously mentioned, after the completion of VFL training, the global model converges, and the sum of the intermediate parameters uploaded by clients $C_A$, $C_B$, and $C_C$ should be approximately zero. Thus, to make the sum of the intermediate parameters of the remaining clients $C_A$ and $C_B$ close to zero after removing the target client $C_C$, it is only necessary to update the model in the reverse direction of the last round of uploading the intermediate parameters from the client $C_C$. This can be achieved by correcting the model using the negative value of the intermediate parameters uploaded by the target client $C$ in the last round, as shown in Figure 3, where $H_U = -H_C = 2900$.

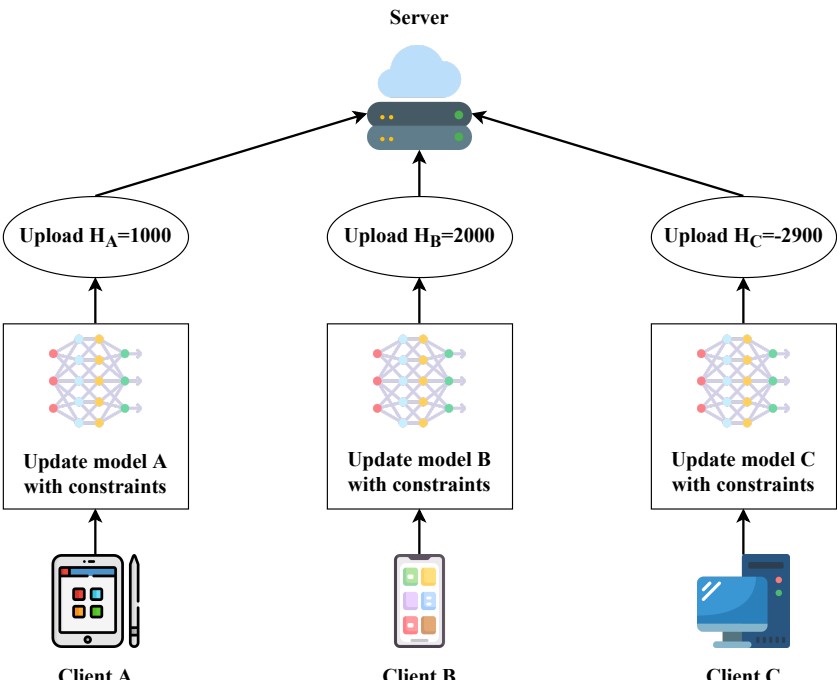

**Figure 3.** Clients upload intermediate parameters $H_i$.

## 5. Experiments

The effectiveness of the proposed unlearning algorithm is evaluated on four datasets using vertical federated logistic regression [34]. The experimental results demonstrate that the proposed algorithm effectively removes the contribution of the target client from the global model. Furthermore, adding intermediate parameter constraints to the client in the training process enables the recovery of the accuracy of the unlearned model with just one extra round of updates.

## 5.1. Datasets Description

Four benchmark datasets are utilized in the experiments, including Cod-RNA [35], Iris [36], Adult [37], and Breast Cancer [38].

Cod-RNAis a dataset used in bioinformatics for predicting the secondary structure of RNA molecules. It contains 59,535 samples, each of which represents a short RNA sequence of length 8. The samples are labeled as belonging to one of two categories, corresponding to two different RNA sequences. The goal is to classify new RNA sequences into the correct category based on their secondary structure.

Iris is a well-known dataset used for classification tasks. It contains 150 samples of iris flowers, each of which has four features: sepal length, sepal width, petal length, and petal width. The samples are labeled as belonging to one of three categories of iris flower: Setosa, Versicolor, or Virginia. The dataset is often used to test the effectiveness of classification algorithms.

Adult is a dataset extracted from the 1994 US Census database, containing information about 48,842 individuals and their income levels. The dataset is divided into 32,561 training samples and 16,281 testing samples. Each sample has 14 features, including age, education level, marital status, occupation, and more. The goal is to predict whether an individual's annual income is greater than $50,000.

Breast Cancer is a dataset used for breast cancer diagnosis. It contains 569 samples, each of which represents a patient with breast cancer. The samples have ten features related to the size, shape, and texture of the cell nuclei in images of the breast tissue. The samples are labeled as either benign or malignant, based on the diagnosis of the patient. The dataset is often used to test the accuracy of machine-learning models for breast cancer diagnosis.

## 5.2. Overview of Experimental Setup

In the VFL setup, a scenario where four clients collaborate to train a vertical federated logistic regression model. An honest server is responsible for aggregating intermediate parameters and distributing public keys. After completing the training process, a random client requests the server to forget its contribution to the global model due to privacy concerns. In response, the server removes the target client from the global model and applies the proposed unlearning algorithm to correct the unlearned model. The training and unlearning processes are both encrypted using the Paillier homomorphic encryption algorithm [39], which guarantees the privacy and security of the client's data.

### 5.2.1. Model Architectures

The vertical federated logistic regression model [40] is employed as the global model in the VFL setup due to its capability to monitor the variation of intermediate parameters uploaded by clients. To restore the accuracy of the unlearned model, the uploaded intermediate parameters from each client are constrained using the loss function presented in Equation (6). This ensures that the intermediate parameters align with the desired accuracy and contributes to the overall improvement of the unlearned model.

To correct the unlearned model after removing the target client's contribution, the server must keep track of the intermediate parameters uploaded by all clients in the last round. This step is crucial as the contributions of the target client are leveraged to expedite the restoration of the prediction accuracy of the unlearned model during the unlearning process. By utilizing the target client's contributions, the model's accuracy can be effectively improved, and its overall performance can be enhanced.

### 5.2.2. Data Processing

To preprocess the four datasets, the features were standardized and normalized. Subsequently, the feature values were divided equally among the four clients to create their respective local datasets. One of the clients was randomly selected to provide the dataset labels for model training.

To test the effectiveness of the unlearning algorithm against data-poisoning attacks, the target client, which held toxic labels, was removed from the training process. To prevent the model from failing to train due to the lack of labels after unlearning the target client, clean labels were assigned to a randomly selected client. This approach ensured the continuity of the training process and facilitated the assessment of the unlearning effect. By adopting this strategy, the model remained stable and capable of learning even after unlearning the target client.

### 5.2.3. Unlearning Target with Data-Poisoning Attacks

To evaluate the effectiveness of vertical federated unlearning, data-poisoning attacks were employed [41]. Specifically, the features of the datasets were maintained unchanged while the labels of a certain percentage of the samples were reversed to create poisoned samples. These poisoned samples were then combined with normal samples to form the training dataset for VFL. The poisoning rate of attacks was set to 3%.

After removing the contribution of the target client, A randomly selected client was assigned clean labels, and the poisoned samples were used to test the unlearned model. The success rate of the data-poisoning attacks was used as a metric to assess whether the unlearned model had successfully forgotten the contribution of the target client. If the success rate fell below a certain threshold, it indicated that the unlearning process was successful.

### 5.3. Unlearned Model Performance Evaluation

In this subsection, the performance of the unlearned model is evaluated under different scenarios. First, The prediction accuracy and data-poisoning attack success rate of the model trained without using the unlearning method but directly removing the target client is examined. As shown in Figure 4, the global model exhibits relatively high prediction accuracy and data-poisoning attack success rate after VFL training is completed. However, the accuracy of the global model decreases significantly after forgetting the contribution of the target client. To recover the prediction accuracy of the unlearned model, further training of the model with the remaining clients' datasets is necessary.

Next, the performance of the proposed unlearning method is demonstrated in Figure 4. The "unlearn" lines show the proposed unlearning method. The "compare" lines represent the method that directly removes the target clients and continues training. The "retrain" lines represent the traditional retraining from scratch method. The "acc" lines stand for the accuracy of the model on the test datasets. The "atk succ" lines represent the success rate of data-poisoning attacks. It was observed that the prediction accuracy of the unlearned model showed significant improvement when corrected with the intermediate parameters uploaded by the target client in the last round. Compared to retraining from scratch, the method requires only one round of updates, which leads to a dramatic improvement in training efficiency. Furthermore, the impact of data-poisoning attacks on the original global model would not transfer to the unlearned model because the unlearned model does not use any data from the target client. Therefore, it is immune to data-poisoning attacks from the original global model. This demonstrates the privacy protection of the vertical federated unlearned model for the target client since the effect of client contribution will be completely removed from the unlearned model.

In Table 1, the results of the proposed unlearning method on different datasets are presented. The "Training" row shows the performance of the vertical federated logistic regression model after training with toxic labels and a loss function with intermediate parameter constraints. As can be observed, the success rate of poisoning attacks on the three datasets is high, indicating that the model has been affected by the target client's attack. However, the Adult dataset is insensitive to such attacks, so the change in the success rate of data-poisoning attacks after unlearning is not significant. The "Unlearning" row shows the performance of the model after removing the target client using the proposed unlearning algorithm. Compared to the "Training" row on the four datasets, the prediction accuracy of

the unlearned model decreases slightly, but the success rate of data-poisoning attacks drops significantly. This shows that the unlearned model successfully forgets the contribution of the target client while maintaining a reasonable level of prediction accuracy. The "Post-Training" row reports the performance that the unlearned model achieves by continuing training after forgetting the contribution of the target client. From the experimental results on the four datasets, the success rate of data-poisoning attacks on the "Post-training" model is observed to have slightly increased compared to the unlearned model. This suggests that although the unlearned model appears to have eliminated the influence of the target client, it may still potentially retain some contributions from the target client, which are exposed after "Post-training". The "Re-Training" row reports the results of removing the target client data from the test datasets and retraining from scratch, which is the least efficient method. Retraining can also completely remove the poisoning effect from the target client, but it requires a significant amount of training rounds. The "R/U" row compares the performance of the unlearning model with that of the retraining model. As observed, the test accuracy and the success rate of data-poisoning attacks on the unlearning model are slightly lower than those of the retraining model, but the training rounds are significantly reduced, demonstrating the advantage of the proposed unlearning method.

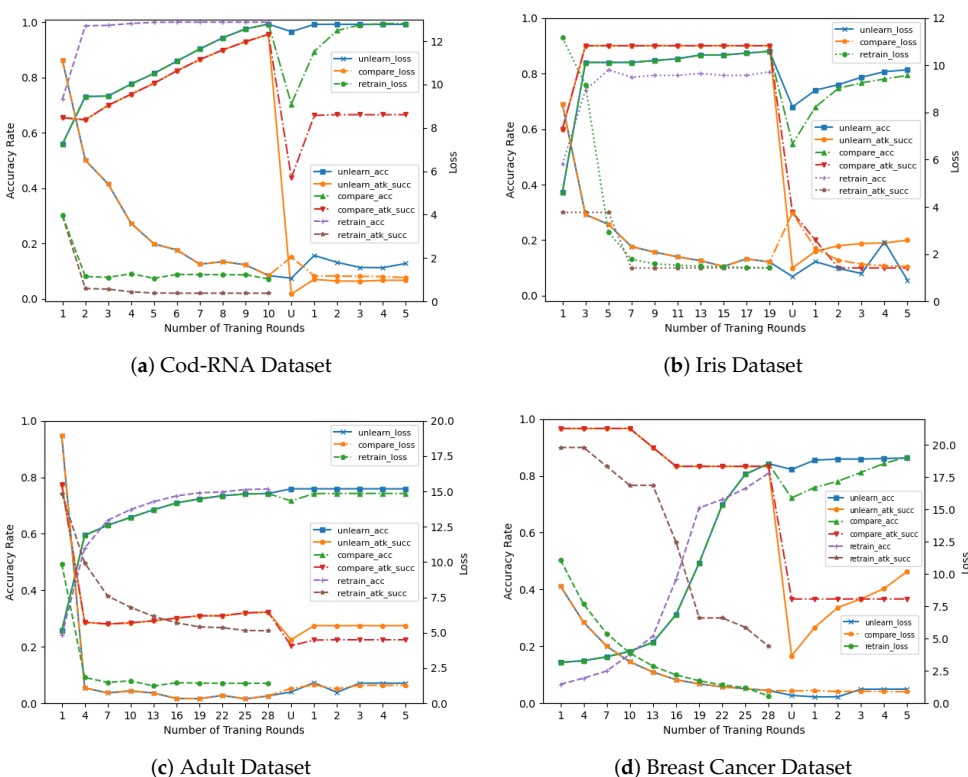

**Figure 4.** The performance of the unlearning method with training rounds is shown on four datasets. The "U" on the horizontal axis shows the performance of the model after unlearning.

From the results, it can be concluded that removing the contribution of the target client using this unlearning algorithm can effectively eliminate its influence on the unlearned model. The success rate of data-poisoning attacks on the unlearned model is significantly reduced compared to the original global model on all datasets. In addition, the proposed method of imposing constraints on the intermediate parameters can help reduce the deviations in model accuracy caused by parameter subtraction.

However, according to the experimental findings, even though the unlearned model has eliminated the influence of the target client, it may still potentially retain some contributions from the target client, and this latent information may be revealed as the model is

post-trained. Therefore, eliminating the potential contributions of the target client from the model is still a research question worthy of investigation.

**Table 1.** The performance of the proposed unlearning method and its efficiency improvement is shown.

| Datasets | Cod-RNA | | | Iris | | |
|---|---|---|---|---|---|---|
| | Test Acc | Atk Succ | Train Round | Test Acc | Atk Succ | Train Round |
| Training | 98.28% | 95.60% | 10 | 84.66% | 90% | 20 |
| Unlearning | 96.56% | 1.46% | 1 | 68.00% | 10% | 1 |
| Post-Training | 99.20% | 34.66% | 5 | 81.33% | 20% | 5 |
| Re-Training | 99.29% | 1.66% | 5 | 82.00% | 15% | 9 |
| R/U | 1.02× | 1.13× | 5.00× | 1.20× | 1.50× | 9.00× |
| Datasets | Adult | | | Breast Cancer | | |
| | Test Acc | Atk Succ | Train Round | Test Acc | Atk Succ | Train Round |
| Training | 75.28% | 28.20% | 30 | 89.20% | 86.66% | 20 |
| Unlearning | 72.91% | 22.49% | 1 | 82.25% | 16.66% | 1 |
| Post-Training | 74.52% | 27.70% | 5 | 89.66% | 46.33% | 5 |
| Re-Training | 75.90% | 25.70% | 20 | 85.93% | 20.00% | 20 |
| R/U | 1.04× | 1.14× | 20× | 1.04× | 1.20× | 20× |

*5.4. Ablation Experiment*

In this section, the effectiveness of the proposed vertical federated unlearning algorithm is assessed through ablation experiments. The aim of these experiments is to examine the individual contributions of different components in the algorithm and their impact on the overall performance. Three distinct scenarios are compared:

Direct removal of target client: In this scenario, the target client is directly removed from the training process, and the resulting performance of the model is assessed. The purpose is to evaluate the impact of removing the target client on the overall performance of the model.

Constraint-based approach for intermediate parameters: In this scenario, constraints are imposed on the intermediate parameters during the training phase, followed by the removal of the target client. The performance of the model is then evaluated under these constraints. This analysis helps determine the effectiveness of incorporating intermediate parameter constraints in improving the model's performance.

Updating other clients with target client's intermediate parameters: In this scenario, the other clients are updated with the intermediate parameters obtained from the last round of uploading by the target client, after the target client has been removed. The model's performance is examined following this update. The purpose is to investigate the influence of incorporating the target client's information on the performance of the model.

By conducting these ablation experiments and comparing the outcomes of each scenario, insights can be gained into the effectiveness of each component and its impact on the overall performance of the vertical federated unlearning algorithm. This analysis helps in understanding the contributions of different elements in achieving successful unlearning and improving the robustness of the model.

The experimental results presented in Table 2 provide insights into the performance of different methods in the unlearning process. The analysis of these results reveals several significant findings. First, the method of directly removing the target client has a notable impact on the prediction accuracy of the unlearned model. Moreover, this method exhibits a high success rate of poisoning attacks, indicating its inability to effectively eliminate the target client's contribution to the VFL model. On the other hand, the methods involving constraints on intermediate parameters and subtracting target client updates demonstrate improved performance. These methods result in higher prediction accuracy and lower success rates of poisoning attacks compared to direct removal of the target client. Lastly, the proposed method combines the strengths of these approaches by incorporating constraints

on intermediate parameters and subtracting target client updates, leading to high prediction accuracy and the lowest success rate of poisoning attacks.

**Table 2.** The performance of the proposed unlearning algorithm in ablation experiments is compared to three other methods.

| Methods | Cod-RNA | | Iris | |
| --- | --- | --- | --- | --- |
| | Test Acc | Atk Acc | Test Acc | Atk Acc |
| Directly remove target client | 71.68% | 42.66% | 56.00% | 28.00% |
| Constrain intermediate parameters | 73.40% | 34.25% | 57.33% | 24.00% |
| Subtract target client updates | 94.42% | 3.84% | 68.67% | 18.00% |
| Our Methods (Constrain + Subtract) | 96.56% | 1.46% | 76.00% | 10.00% |
| Methods | Adult | | Breast Cancer | |
| | Test Acc | Atk Acc | Test Acc | Atk Acc |
| Directly remove target client | 68.60% | 29.32% | 70.89% | 38.75% |
| Constrain intermediate parameters | 69.70% | 28.45% | 72.32% | 35.00% |
| Subtract target client updates | 72.91% | 26.82% | 76.92% | 19.92% |
| Our Methods (Constrain + Subtract) | 78.46% | 22.49% | 82.25% | 16.66% |

In conclusion, the experimental results underscore the effectiveness of the proposed unlearning method. A comparison of the methods reveals that the constrained intermediate parameters method exhibits a slight improvement in unlearning effectiveness, while the subtracting target client updates method demonstrates a significant enhancement in prior learning effectiveness. By combining the advantages of these two methods, the proposed approach achieves more favorable outcomes. These ablation experiments serve as compelling evidence of the effectiveness of the method and shed light on the influence of each design component on unlearning performance. The insights gained from these results provide a solid foundation for future research on vertical federated unlearning.

*5.5. Robustness Experiment*

Robustness experiments play a crucial role in evaluating the effectiveness of vertical federated unlearning. These experiments allow us to assess the stability and reliability of unlearned models when faced with various challenges and attacks. In the context of vertical federated unlearning, the model must continuously forget the contributions of specific clients while still retaining its ability to learn from data provided by other clients. By conducting robustness experiments under different scenarios, the robustness of the algorithm and its ability to adapt to changes in data distribution can be evaluated. This provides valuable insights into the performance of unlearned models in real-world scenarios where the distribution of data among clients may vary.

In the experiment, the robustness of the unlearned model is assessed using data distribution change attacks. Specifically, a scenario is considered where a VFL model is trained collaboratively by four clients, with one client being designated as the target client. The effectiveness of unlearning is evaluated in three different scenarios:

Even Distribution: In this scenario, the dataset features are evenly divided among all clients, with each client holding an equal share of 25% of the features.

More Data on Target Client: In this scenario, a larger proportion of the dataset features (40%) is allocated to the target client, while the other clients retain their respective shares of 20% of the features.

Fewer Data on Target Client: In this scenario, a smaller proportion of the dataset features (10%) is allocated to the target client, while the remaining clients still hold their shares of 30% of the features.

Due to the limited number of features in the Iris dataset, it is not well-suited for conducting robustness experiments involving changes in data distribution. To address this limitation, experiments were conducted on three additional datasets: Cod-RNA, Adult, and Breast Cancer. These datasets provide a more diverse range of features and enable us

to evaluate the robustness of the algorithm under different data distribution scenarios. The experimental results on these datasets are presented in Table 3, which provides insights into the performance of the algorithm in terms of robustness and adaptability to varying data distributions.

**Table 3.** Experimental results on the robustness of the proposed unlearning method under different data distributions.

| Data Distribution | Cod-RNA | | Adult | | Breast Cancer | |
|---|---|---|---|---|---|---|
| | Test Acc | Atk Acc | Test Acc | Atk Acc | Test Acc | Atk Acc |
| Even Distribution | 96.56% | 1.46% | 78.46% | 22.49% | 82.25% | 16.66% |
| Target Client (40%) | 93.76% | 7.54% | 74.91% | 26.90% | 68.75% | 27.85% |
| Target Client (10%) | 97.42% | 1.28% | 79.22% | 20.21% | 90.42% | 10.00% |

Based on the experimental results, it can be concluded that the performance of the unlearned model is negatively affected when the data distribution is concentrated on the target client. This can be attributed to the fact that the original model learned a significant amount of data from the target client, making it more challenging to completely forget the impact of these data. Consequently, the prediction accuracy of the unlearned model is significantly affected. On the other hand, when there is less data distributed to the target client, the unlearned model performs better. In this case, the original model only relies on a smaller portion of the target client's data, making it easier to forget the contributions of this client. As a result, the unlearned model achieves higher prediction accuracy.

These findings highlight the importance of considering the data distribution in the context of vertical federated unlearning. It is crucial to strike a balance between the contributions of different clients and ensure that the unlearning process effectively removes the impact of specific clients without significantly compromising the overall model performance. The insights gained from these robustness experiments can guide the development of more robust unlearning algorithms and contribute to the advancement of vertical federated unlearning. Future research directions may include exploring techniques for handling non-IID (non-identically distributed) data distributions and developing strategies to improve the resilience of unlearned models against varying data distributions.

## 6. Practical and Theoretical Implications

The practical implications of vertical federated unlearning are significant in the field of privacy-preserving machine learning. By introducing a mechanism for selectively forgetting specific client contributions while preserving knowledge from other clients, this method ensures data privacy and confidentiality. It finds practical applications in various industries and domains where federated learning is used to leverage distributed data while safeguarding individual data privacy.

Moreover, the adaptability of the vertical federated unlearning method has practical benefits in scenarios where user data characteristics change over time or where outdated or irrelevant information needs to be removed. The dynamic adjustment of the model's memory allows it to effectively adapt to evolving data distributions and improve performance on more recent data. Inspired by the practical and theoretical significance of Chen et al. [42] and Yuan et al. [43], this method has practical implications in domains such as healthcare, finance, and telecommunications, where data characteristics may exhibit temporal variations.

From a theoretical perspective, the proposed vertical federated unlearning method contributes to the field of federated learning by addressing the challenges associated with unlearning in vertical federated settings. It advances the understanding of how models can selectively forget specific client contributions while retaining knowledge from other clients. This theoretical advancement presents new research avenues in federated learning and contributes to the development of more robust and flexible learning algorithms.

Additionally, the vertical federated unlearning method provides insights into the dynamics of VFL in distributed environments. By studying the unlearning process within a vertical federated context, a deeper understanding of how models can evolve over time and adapt to changing data distributions is achieved. This contributes to the theoretical foundations of VFL and provides insights into the mechanisms of model adaptation in distributed settings.

In conclusion, the practical and theoretical implications of the proposed vertical federated unlearning method encompass the preservation of data privacy in federated learning, adaptability to changing data distributions, and advancements in the understanding of VFL in distributed environments. This innovative approach has the potential to transform the field of federated learning and drive future research in privacy-preserving machine-learning techniques.

## 7. Conclusions and Future Work

In this paper, the challenges of vertical federated unlearning have been addressed, and a novel algorithm has been proposed to tackle these challenges. The proposed algorithm combines constraints on intermediate parameters and the subtraction of target client updates to enhance the performance of the unlearned model. Experimental results on benchmark datasets have demonstrated the effectiveness of the proposed unlearning algorithm. However, it is important to acknowledge that the unlearned model may still potentially retain information from the target clients, even if it is not immediately apparent. This potential information leakage could manifest itself through post-training analysis. Further research is needed to understand and mitigate this potential information leakage.

Moving forward, there are several promising directions for future research in the field of vertical federated unlearning. First, investigating the phenomenon of potential information retention in the unlearned model and developing strategies to address it will be crucial to enhance the privacy guarantees and security of the unlearning process. Second, robustness experiments have indicated that the proposed method is sensitive to changes in data distribution, particularly in non-IID scenarios. Exploring techniques to improve the performance and applicability of vertical federated unlearning in non-IID settings will be an important area of investigation. By addressing these challenges, the field of vertical federated unlearning can be further advanced, contributing to the development of more robust and privacy-preserving machine-learning techniques.

**Author Contributions:** Conceptualization, Z.D. and C.M.; methodology, Z.D. and C.M.; software, Z.D. and Z.H.; validation, M.D. and L.Y.; formal analysis, Z.H. and C.M.; investigation, M.D. and L.Y.; resources, Z.D.; data curation, Z.H.; writing—original draft preparation, Z.D.; writing—review and editing, C.M. and M.D.; visualization, Z.D. and Z.H.; supervision, C.G. and Z.L.; project administration, Z.L.; funding acquisition, C.G. All authors have read and agreed to the published version of the manuscript.

**Funding:** This work was supported by the National Key R&D Program of China (2021YFB3100700), the National Natural Science Foundation of China (62076125, 62032025, U20B2049, U20B2050, U21A20467, 61702236, 6207020639), the Key R&D Program of Guangdong Province (2020B0101090002), the Shenzhen Science and Technology Program (JCYJ202103324134810028, JCYJ202103324134408023), the Ministry of Industry and Information Technology of China (TC200H02X), and the Shenzhen Virtual University Park Support Scheme (YFJGJS1.0), the Guangdong Basic and Applied Basic Research Foundation (2021A1515012650), the National Natural Science Foundation of China under Grant No. 62002170, the Youth Foundation Project of Zhejiang Lab (No. K2023PD0AA01). Long Yuan is supported by National Key RD Program of China 2022YFF0712100.

**Data Availability Statement:** The data presented in this study are openly available in the public repository. https://github.com/dateaaalive/vfl.

**Conflicts of Interest:** The authors declare no conflict of interest.

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
