# Peer review of "Vertical Federated Unlearning on the Logistic Regression Model"

_electronics, doi:10.3390/electronics12143182_

Round 1

Reviewer 1 Report

In this paper, the author propose a vertical federated unlearning method by imposing constraints on intermediate parameters and subtracting target client updates from the global model. To evaluate the performance of their method, they also develop a poisoning attack. The topic is quite innovative. I suggest the author to make a minor revision. After reading this paper, I have following suggestions:

1st  The author adopt a logistic regression model as a base model for the classification task. In reality, the logistic is prone to make very false decision if there exist an amount of noise. I suggest the author give some discussion about the robustness of such method.

2nd  The author should add a section of ablation study to discuss the impact of each parameters toward the certain method.

3rd  The related work need to be enriched. I suggest the author to discuss the poisoning attack by reading the following papers:

3.1 Tolpegin, V., Truex, S., Gursoy, M. E., & Liu, L. (2020). Data poisoning attacks against federated learning systems. In Computer Security–ESORICS 2020: 25th European Symposium on Research in Computer Security, ESORICS 2020, Guildford, UK, September 14–18, 2020, Proceedings, Part I 25 (pp. 480-501). Springer International Publishing.

      3.2 Fang, Minghong, Xiaoyu Cao, Jinyuan Jia, and Neil Gong. "Local model poisoning attacks to {Byzantine-Robust} federated learning." In 29th USENIX security symposium (USENIX Security 20), pp. 1605-1622. 2020.

       3,3 Fung, Clement, Chris JM Yoon, and Ivan Beschastnikh. "Mitigating sybils in federated learning poisoning." arXiv preprint arXiv:1808.04866 (2018).

       3.4  Zhou, Xingchen, Ming Xu, Yiming Wu, and Ning Zheng. "Deep model poisoning attack on federated learning." Future Internet 13, no. 3 (2021): 73.

        3.5 Sun, Jingwei, Ang Li, Louis DiValentin, Amin Hassanzadeh, Yiran Chen, and Hai Li. "Fl-wbc: Enhancing robustness against model poisoning attacks in federated learning from a client perspective." Advances in Neural Information Processing Systems 34 (2021): 12613-12624.

        3.6 Chen, Ziheng, Fabrizio Silvestri, Jia Wang, Yongfeng Zhang, and Gabriele Tolomei. "The dark side of explanations: Poisoning recommender systems with counterfactual examples." SIGIR (2023).

The english is good, but still need to be polished

Author Response

We appreciate your valuable feedback and insightful comments on our manuscript. We have carefully considered all the suggestions and have made the necessary revisions to improve the quality and clarity of the paper. The overview of the changes is summarized as follows.

Point 1: The author adopt a logistic regression model as a base model for the classification task. In reality, the logistic is prone to make very false decision if there exist an amount of noise. I suggest the author give some discussion about the robustness of such method.

Response 1: Thank you for this important comment. Robustness experiments are indeed crucial for evaluating the resilience and reliability of models in the face of various challenges and disturbances. In section 5.5, we added robustness experiments (page 15, line 546) to assess the algorithm's resilience to changes in data distribution. These experiments revealed that as the data distribution became more concentrated toward the target client, the performance of the unlearned model showed a slight decrease. This observation indicates that the algorithm possesses a certain degree of robustness in handling variations in data distribution.

Point 2: The author should add a section of ablation study to discuss the impact of each parameters toward the certain method.

Response 2: Thank you for this important comment. The ablation experiment is indeed crucial for understanding the role of each module of the algorithm. In section 5.4, we added ablation experiments (page 16, line 591) and compared our methods with three methods. Experiments have shown that compared to these three methods, our proposed unlearning algorithm has better performance.

Point 3: The related work need to be enriched. I suggest the author to discuss the poisoning attack by reading the following papers

Response 3: Thank you for this important comment. References play a crucial role in providing a comprehensive understanding of the background and research status of algorithms. In section 2, we added the literature review on data poisoning attacks (page 4, line 147) in federated learning that you provided. Additionally, we have included an updated literature review on machine unlearning (page 3, line 117) and federated unlearning (page 3, line 133) in 2023. They contribute to the credibility and validity of our work by demonstrating our awareness of the existing literature and highlighting the novel aspects of our approach.

Finally, we would like to thank the reviewers for providing very constructive feedback that is very useful for enhancing the quality of our paper.

Reviewer 2 Report

First of all, I appreciate the opportunity to review the paper Vertical Federated Unlearning on Logistic Regression Model.  The paper deals with very interesting and actual problems.

Suggestions are below:

·        Scientific paper writing without "we"; "our" (only in the introduction you have 13 „we“)

·        The last paragraph in the introduction section is a short structure of the paper (several sentences for each section). This is missing.

·        Section 2 Related works should be a Literature review with much more details. This is a very important section. It is necessary to understand the purpose and aim of the paper as well as its "position" in relation to previous research (also gap analysis).

·        The separate section Practical and theoretical implications (or Discussion) is missing. The existing section Discussion is very modest. This confirms the lack of scientific and practical contributions.

·        The Figure 3 caption is too long.

“The performance of the unlearning method is shown on four datasets. The “unlearn” lines show our proposed unlearning method. The “compare” lines represent the method that directly removes the target clients and continues training. The “retrain” lines represent the traditional retraining from the scratch method. The “acc” lines stand for the accuracy of the model on the test datasets. The “atk succ” lines represent the success rate of data poisoning attacks. The “U” on the horizontal axis shows the performance of the model after unlearning”

·        Conclusion section is not on a satisfactory level. The conclusion in scientific papers is very important.

o   Limitations of your research must be emphasized

o   Future research directions are poorly written.

·        The scientific and practical contributions should be emphasized.

.

Author Response

We appreciate your valuable feedback and insightful comments on our manuscript. We have carefully considered all the suggestions and have made the necessary revisions to improve the quality and clarity of the paper. The overview of the changes is summarized as follows.

Point 1: Scientific paper writing without "we"; "our" (only in the introduction you have 13 „we“)

Response 1: Thank you for this important comment. English grammar is of great importance when writing academic papers. In this paper, an effort has been made to replace instances of the pronouns "we" and "our" with the passive voice. This adjustment helps maintain a more formal and objective tone throughout the manuscript. 

Point 2: The last paragraph in the introduction section is a short structure of the paper (several sentences for each section). This is missing.

Response 2: Thank you for this important comment. This has indeed enhanced the clarity of our paper. We added a brief structure of the paper in the last paragraph of the introduction section (page 2, line 88).

Point 3: Section 2 Related works should be a Literature review with much more details. This is a very important section. It is necessary to understand the purpose and aim of the paper as well as its "position" in relation to previous research (also gap analysis).

Response 3: Thank you for this important comment. References play a crucial role in providing a comprehensive understanding of the background and research status of algorithms. In section 2, we have included an updated literature review on machine unlearning (page 3, line 117) and federated unlearning (page 3, line 133) in 2023. Additionally, we have incorporated a literature review on data poisoning attacks in Federated learning (page 4, line 147). They contribute to the credibility and validity of our work by demonstrating our awareness of the existing literature and highlighting the novel aspects of our approach.

Point 4: The separate section Practical and theoretical implications (or Discussion) is missing. The existing section Discussion is very modest. This confirms the lack of scientific and practical contributions.

Response 4: Thank you for this important comment. The practical and theoretical implications play a crucial role in this paper. In section 6, we have included a discussion on the theoretical and practical implications (page 17, line 638) of our proposed algorithm.

Point 5:  The Figure 3 caption is too long. 

Response 5: Thank you for this important comment. This has indeed enhanced the clarity of our paper. We have shortened the title of Figure 3 (page 13).

Point 6: Conclusion section is not on a satisfactory level. The conclusion in scientific papers is very important.

Response 6: Thank you for this important comment. The conclusion of the paper is crucial for providing a concise overview of the key findings and contributions. We have revised the conclusion (page 18, line 671) to highlight the limitations of our method and to provide a comprehensive summary of future research directions.

Point 7: The scientific and practical contributions should be emphasized.

Response 7: Thank you for this important comment. This has indeed enhanced the clarity of our paper. In section 1, we presented the contributions of our work (page 2, line 68). In section 6, we elaborated on the theoretical and practical significance of our research (page 17, line 638).

Finally, we would like to thank the reviewers for providing very constructive feedback that is very useful for enhancing the quality of our paper.

Reviewer 3 Report

The authors claimed they used real life datasets "four real-life datasets demonstrate..." However, those were benchmark datasets not real life. It should be changed to benchmark datasets. Provide a flow diagram to depict the conceptual framework for easy understanding. Repeat the experiment with 5 different data partition ratios e.g. 80:20; 70:30; 50:50; 60:40 to see the robustness of the proposal. each of the experiment should be perfomed multiple number of times say 20, 25, 30 or any number because the study is dealing with intelligent algorithm and compute the mean, sd, best and worst accuracy. Convergence time should be computed and tabulated. The study should clearly mention the algorithm adopted and provide the theoretical background of the algorithms including mathematical background. The compared algorithms should be clearly stated. The literature review is weak strengthen it by adding more studies pointing out the algorithms adopted in each of the study. The study is not up-to-date, 2023 studies are missing, add relevant 2023 studies to show current state of the art

Minor issue

Author Response

We appreciate your valuable feedback and insightful comments on our manuscript. We have carefully considered all the suggestions and have made the necessary revisions to improve the quality and clarity of the paper. The overview of the changes is summarized as follows.

Point 1: The authors claimed they used real life datasets "four real-life datasets demonstrate..." However, those were benchmark datasets not real life. It should be changed to benchmark datasets.

Response 1: Thank you for this important comment. We are indeed using the benchmark dataset, and the “real dataset” in the paper has been replaced with the “benchmark dataset”.

Point 2: Provide a flow diagram to depict the conceptual framework for easy understanding. 

Response 2: Thank you for this important comment. Flowcharts are crucial for understanding our algorithms. A flowchart of the algorithm has been added at the top of page 7 to enhance understanding of our approach.

Point 3: Repeat the experiment with 5 different data partition ratios e.g. 80:20; 70:30; 50:50; 60:40 to see the robustness of the proposal. each of the experiment should be perfomed multiple number of times say 20, 25, 30 or any number because the study is dealing with intelligent algorithm and compute the mean, sd, best and worst accuracy.

Response 3: Thank you for this important comment. Robustness experiments are indeed crucial for evaluating the resilience and reliability of models in the face of various challenges and disturbances. In section 5.5, we added robustness experiments (page 15, line 546) to assess the algorithm's resilience to changes in data distribution. These experiments revealed that as the data distribution became more concentrated toward the target client, the performance of the unlearned model showed a slight decrease. This observation indicates that the algorithm possesses a certain degree of robustness in handling variations in data distribution.

Point 4: The study should clearly mention the algorithm adopted and provide the theoretical background of the algorithms including mathematical background. 

Response 4: Thank you for this important comment. The description of the algorithm background is crucial for providing the necessary context to understand the algorithm. In section 3, we have introduced the background of the algorithm (page 4, line 170) and highlighted the challenges of vertical federated unlearning. 

Point 5: The compared algorithms should be clearly stated. 

Response 5: Thank you for this important comment. The ablation experiment is indeed crucial for understanding the role of each module of the algorithm. In section 5.4, we added ablation experiments (page 16, line 591) and compared our methods with three methods. Experiments have shown that compared to these three methods, our proposed unlearning algorithm has better performance.

Point 6: The literature review is weak strengthen it by adding more studies pointing out the algorithms adopted in each of the study. The study is not up-to-date, 2023 studies are missing, add relevant 2023 studies to show current state of the art.

Response 6: Thank you for this important comment. References play a crucial role in providing a comprehensive understanding of the background and research status of algorithms. In section 2, we have included an updated literature review on machine unlearning (page 3, line 117) and federated unlearning (page 3, line 133) in 2023. Additionally, we have incorporated a literature review on data poisoning attacks in Federated learning (page 4, line 147). They contribute to the credibility and validity of our work by demonstrating our awareness of the existing literature and highlighting the novel aspects of our approach.

Finally, we would like to thank the reviewers for providing very constructive feedback that is very useful for enhancing the quality of our paper.

Round 2

Reviewer 2 Report

The paper is better after corrections. Some additaional explanations.

Point 1: Scientific paper writing without "we"; "our" (only in the introduction you have 13 „we“)

Response 1: Thank you for this important comment. English grammar is of great importance when writing academic papers. In this paper, an effort has been made to replace instances of the pronouns "we" and "our" with the passive voice. This adjustment helps maintain a more formal and objective tone throughout the manuscript. 

MY SUGGESTION IS TO AVOID SCIENTIFIC PAPER WRITING IN THE “FIRST PERSON”. (still exist: line 11, line 30, etc)

Pont 2. Additional comment for Table 1 caption is same as the previous comment for Figure 3 (previous round)

After these corrections, the paper should be accepted

.

Author Response

We appreciate your valuable feedback and insightful comments on our manuscript. We have carefully considered all the suggestions and have made the necessary revisions to improve the quality and clarity of the paper. The overview of the changes is summarized as follows.

Point 1: Scientific paper writing without "we"; "our" (only in the introduction you have 13 „we“)

Response 1: Thank you for this important comment. We have revised the scientific paper to ensure that all “first person” references have been replaced. Now the paper does not contain any "we" and "our".

Pont 2. Additional comment for Table 1 caption is same as the previous comment for Figure 3 (previous round)

Response 2: Thank you for this important comment. We have shortened the title of Table 1 (page 15) and rechecked the additional comment of all tables and figures.

Finally, we would like to thank the reviewers for providing very constructive feedback that is very useful for enhancing the quality of our paper.

Reviewer 3 Report

All issues resolved

Good but little minor edit before publication

Author Response

We appreciate your valuable feedback and insightful comments on our manuscript. We have carefully considered all the suggestions and have made the necessary revisions to improve the quality and clarity of the paper.